# Diversity and Toxigenicity of Fungi that Cause Pineapple Fruitlet Core Rot

**DOI:** 10.3390/toxins12050339

**Published:** 2020-05-21

**Authors:** Bastien Barral, Marc Chillet, Anna Doizy, Maeva Grassi, Laetitia Ragot, Mathieu Léchaudel, Noel Durand, Lindy Joy Rose, Altus Viljoen, Sabine Schorr-Galindo

**Affiliations:** 1Qualisud, Université de Montpellier, CIRAD, Montpellier SupAgro, Univ d’Avignon, Univ de La Reunion, F-34398 Montpellier, France; marc.chillet@cirad.fr (M.C.); grassimaeva@gmail.com (M.G.); laetitiaragot@ymail.com (L.R.); mathieu.lechaudel@cirad.fr (M.L.); noel.durand@cirad.fr (N.D.); sabine.galindo@umontpellier.fr (S.S.-G.); 2CIRAD, UMR Qualisud, F-97410 Saint-Pierre, Reunion, France; 3CIRAD, UMR PVBMT, F-97410 Saint-Pierre, Reunion, France; doizy.anna@gmail.com; 4CIRAD, UMR Qualisud, F-97130 Capesterre-Belle-Eau, Guadeloupe, France; 5CIRAD, UMR Qualisud, F-34398 Montpellier, France; 6Department of Plant Pathology, Stellenbosch University, Private Bag X1, Matieland 7600, South Africa; lindym@sun.ac.za (L.J.R.); altus@sun.ac.za (A.V.)

**Keywords:** *Fusarium ananatum*, *Talaromyces stollii*, beauvericin, fumonisin, *Ananas comosus*

## Abstract

The identity of the fungi responsible for fruitlet core rot (FCR) disease in pineapple has been the subject of investigation for some time. This study describes the diversity and toxigenic potential of fungal species causing FCR in La Reunion, an island in the Indian Ocean. One-hundred-and-fifty fungal isolates were obtained from infected and healthy fruitlets on Reunion Island and exclusively correspond to two genera of fungi: Fusarium and Talaromyces. The genus Fusarium made up 79% of the isolates, including 108 *F. ananatum*, 10 *F. oxysporum*, and one *F. proliferatum*. The genus Talaromyces accounted for 21% of the isolated fungi, which were all *Talaromyces stollii*. As the isolated fungal strains are potentially mycotoxigenic, identification and quantification of mycotoxins were carried out on naturally or artificially infected diseased fruits and under in vitro cultures of potential toxigenic isolates. Fumonisins B_1_ and B_2_ (FB_1_-FB_2_) and beauvericin (BEA) were found in infected fruitlets of pineapple and in the culture media of Fusarium species. Regarding the induction of mycotoxin in vitro, *F.*
*proliferatum* produced 182 mg kg⁻^1^ of FB_1_ and *F. oxysporum* produced 192 mg kg⁻^1^ of BEA. These results provide a better understanding of the causal agents of FCR and their potential risk to pineapple consumers.

## 1. Introduction

Fruitlet core rot affects pineapple fruit until it becomes unfit for consumption. The earliest reference to investigations of fruitlet core rot (FCR) in pineapples dates back to 1898 in Australia [1]. The symptoms were described as “well-defined dark brown markings immediately beneath the surface, and passing inwards to a depth of ¼” to ½—the malady commencing in separate fruitlets, the central core of the fruit remaining quite healthy”. This description agrees with the symptoms referred to as “black spot”.

*Penicillium* sp. Link was isolated from blossom cups as well as infected fruitlets on the cultivar “Smooth Cayenne” [1]. Since this first description, numerous attempts have been made to identify the causal pathogens. In the first half of the 20th century, the majority of investigators around the world attributed the disease to *Penicillium* sp. [2,3,4,5]. *Penicillium funiculosum* belongs to the subgenus *Biverticillium*. Phylogenetic analysis revealed that the Penicillium subgenera Biverticillium and Talaromyces form a monophyletic clade. With the recent adoption of the “one fungus, one name” concept, the Penicillium subgenus Biverticillium was transferred to Talaromyces. *Penicillium funiculosum* was renamed *Talaromyces funiculosus* [6,7].

In the same period, other studies associated the disease with the Fusarium genus. In Hawaii, a brown discoloration in pineapples fruitlets of the “Smooth Cayenne” cultivar was attributed to the genus *Fusarium* [8]. Fusariosis caused by *Fusarium guttiforme* (formerly *Fusarium moniliforme* var. *subglutinans*, *Fusarium subglutinans*) appeared in Brazil in the 1960s and became the major concern for producers in that region [9,10,11,12,13]. The name “*Fusarium guttiforme*” brought about confusion, and some articles dealing with FCR used it to describe the pathogen [14,15]. More recently, molecular tools led to the identification of *F. ananatum* as responsible for FCR and to differentiate it from *F. guttiforme*, the agent responsible for pineapple fusariosis [16]. *Fusarium ananatum* was also identified on commercial pineapples from Costa Rica, Ecuador, Hawaii, and Honduras [17]. This new species was later observed on pineapple in China [18]. The gene encoding translation elongation factor 1 alpha (TEF-1α) was amplified and sequenced to confirm the identity of the causal fungus in the previous studies. TEF-1α is commonly used to infer phylogenetic relationships between closely related genotypes, particularly in the genus Fusarium [19,20,21].

Finally, some studies associated FCR with a combination of Penicillium and Fusarium [22,23,24]. In Queensland, *T. funiculosus* and *F. verticillioides* were, respectively, recovered from 67% and 25% of isolations from pineapple [25]. An FCR control program conducted in Hawaii showed a predominance of Talaromyces [26]. In South Africa, Talaromyces was isolated in all parts of the fruit, while Fusarium was found only in deep lesions. *Talaromyces funiculosus* was considered the primary causal agent of FCR but later the situation reversed, with a majority of Fusarium observed in pineapples [14,15].

FCR is widespread in all pineapple-producing regions but different fungal communities are associated with the disease [27,28]. Obtaining a better knowledge of the causal pathogen is an essential step in the development of a disease management strategy.

Moreover, the fungal contaminants isolated from pineapples in various studies are possible producers of mycotoxins, such as species of the genus Fusarium. The fumonisins B_1_, B_2_, and B_3_ (FB_1,_ FB_2_, and FB_3_), moniliformin (MON), and beauvericin (BEA) were found in pineapple juice and the skin of commercial pineapple [17]. The *Fusarium* spp. isolated from pineapple fruits, especially *F. ananatum* and *F. proliferatum*, were able to produce those mycotoxins in vitro. *Fusarium oxysporum* was found to produce MON and BEA only. *Fusarium proliferatum* was also described as a producer of fumonisins in different studies [19]. *Fusarium verticillioides* (syn. *F. moniliforme*), one of the two fungi historically associated with FCR, is one of the producers of mycotoxins in cereals, especially fumonisins in wheat [29]. *Penicillium funiculosum*, renamed *Talaromyces funiculosus*, often associated with the FCR disease [24,30,31], was found to produce patulin, especially in fruit [32,33] and sorghum [34].

The early symptoms of FCR are a browning of the flesh under the blossom cup of mature fruit. The black spot can spread to the core but remains confined to the fruitlet. This browning is the result of the oxidation of phenolic compounds into quinones by the enzymes polyphenol oxidase and laccase [35]. Soluble and cell wall-bound phenolic acids are found in the pineapple fruit parenchyma [36,37] and increase dramatically after the onset of the first symptoms of FCR [38,39]. These compounds provide antifungal activity on several species of fungi. Coumaric and ferulic acids inhibit the in vitro mycelial growth of *F. ananatum* and slow the in vivo development of the fungus by co-inoculation in the fruit [40]. In addition to their antifungal properties, some of these compounds interfere with the accumulation of mycotoxins. A previous report indicated that ferulic acid supplementation led to a reduction in mycotoxin production by *F. graminearum* [41]. Phenylpropanoids have the capacity to reduce mycotoxin produced by *Fusarium* spp. in cereals into less toxic compounds and to inhibit toxin biosynthesis [42,43]. In practice, the application of phenolic-rich plant extract and tannic acid allows the control of Fusarium head blight in wheat and prevents mycotoxin contamination [44]. However, hydroxycinnamic acids can also increase the mycotoxin accumulation [45].

In order to establish the safety of the diseased fruit, it is essential (i) to know which species of fungi are the causal agents of the pineapple black spot disease and in what proportions, (ii) to verify whether the fungi are capable of producing mycotoxins in vitro and especially in vivo, and (iii) to determine whether pineapple produces key metabolites or enzymes that are able to inhibit this toxigenicity.

## 2. Results

### 2.1. Morphological Identification

One-hundred-and-fifty fungal isolates were collected from pineapple fruits in Reunion Island and classified into in two genera based on their morphological comparison (Figure 1). The 119 isolates of the Fusarium genus were characterized by fusiform macroconidia, usually with 2–3 septa on carnation leaf agar medium and formed two distinguishable groups; the first group of 108 strains had saffron-colored colonies and the second group had white to pale violet-colored colonies on potato dextrose agar (PDA). The 31 isolates belonging to Talaromyces were characterized by fast growth and colonies with floccose or loosely funiculose textures on PDA, with white, green, and red mycelia. Conidiophores were biverticillate with acerose phialides. The conidial shape was ellipsoidal (2.5–4 × 2–2.5 μm).

Both genera of pathogenic fungi could be found on the same fruit. The morphological data are supported by the phylogenetic results, as discussed below.

### 2.2. Distribution of Isolated Fungi

The genus Fusarium represented 79% of the isolates, and the genus Talaromyces accounted for 21% of the isolates. Fusarium isolates were collected from fruits grown at an average altitude of 310 m, while Talaromyces isolates were collected from fruits grown at an average altitude of 399 m. Figure 2 shows the distribution of locations in Reunion Island from which fungi were isolated. The Fusarium isolates were found in all production areas, as shown on the map. The Talaromyces isolates were found in the eastern and southern regions, which are also the rainiest parts of the island.

### 2.3. Phylogenetic Analysis

TEF analysis was used in this study to define relationships among the isolates belonging to the Fusarium clade. The isolates were compared to the reference sequences deposited in the NCBI GenBank using BLASTN [46]. The aligned dataset, which includes a subset of species classified in Fusarium, had a total length of 709 bp. The model TN93 with a gamma distribution (+G) was the most suitable model for the ML distribution [47]. *Neocosmospora solani* (syn. F. solani) was selected as the outgroup (DQ247710 TEF barcode) [48,49]. Three species of Fusarium were identified: *F. ananatum, F. oxysporum*, and *F. proliferatum* (Appendix A). In the majority of cases, the identification corresponded to *Fusarium ananatum* (108 isolates), followed by *F. oxysporum* (10 isolates), and *F. proliferatum* (one isolate). For example, the selected strains clp028, clp101, clp103, and clp117 showed 100% sequence identity to the *F. ananatum* type strain DQ282171 and DQ282171 [16], confirming the phylogenetic position of this strain in the TEF tree. *Fusarium ananatum* strains clp152, clp153, and clp154 isolated from healthy blossom cups ranked in the same clade as strains isolated from infected fruitlets. *Fusarium oxysporum* (JF740878 TEF barcode) [50] showed 99% sequence identity with selected isolates clp076, clp105, and clp003. *Fusarium verticillioides* (KT716244 TEF barcode) [51] and *F. guttiforme* (DQ282170, DQ282166, and DQ282165) [16] were distinct clades of *F. ananatum.* Only a single occurrence of *F. proliferatum* was observed: the clp081 sequence had a 99% match with AF160280 and AF336913 type strain sequences [52,53].

The Talaromyces genus accounted for 21% of the isolates, and among these 31 strains, five were used in phylogenetic analysis (Appendix A). The ITS gene region defined relationships within Talaromyces. The aligned dataset had a total length of 709 bp. The model TN93 was the most suitable model for the ML distribution. *Talaromyces dendriticus* was selected as the outgroup (JX091486 ITS barcode) [6]. Based on the ITS dataset, the isolates from infected pineapple grouped together with *T. stollii* (JX315670 and JX315676 ITS barcode) [54] type strains formed a distinct clade from *T. funiculosus* (KM066193, KM066195 ITS barcode) [7] type strains with 99% bootstrap support. Some differences were found in the sequences of the Reunion pineapple isolates, which most likely indicates genetic variation in this population.

### 2.4. Mycotoxins Synthesized In Vitro

The ability of the isolates to produce mycotoxins was quantified on a pineapple medium (Table 1). This culture medium was selected to mimic natural conditions. Of the 141 fungi tested, 12 were producers of mycotoxins, all of which belonged to the Fusarium genus. Nine strains of *F. ananatum* were producers of fumonisin B_1_ and B_2_, up to 0.13 and 0.24 mg kg⁻^1^, respectively. *Fusarium oxysporum* produced up to 36.9 mg kg⁻^1^ beauvericin (BEA). The *F. proliferatum* isolate was able to produce the highest concentrations of FB_1_, FB_2_, and BEA of any of the isolates for each type of mycotoxin, with 96.7 mg kg⁻^1^ FB_1_, 11.2 mg kg⁻^1^ FB_2_, and 103.2 mg kg⁻^1^ BEA.

The pineapple juice agar (PJA) medium was enriched with phenolic compounds or enzymatic complexes at 0.5 g L⁻^1^ (Figure 3). This addition resulted in a reduction in toxin accumulation compared to the control for the *F. proliferatum* isolate. Caffeoylquinic acid was the least effective compound for reducing the level of mycotoxins and actually promoted the accumulation of toxins compared to the control for the *F. oxysporum* strain. However, all other compounds decreased the levels of mycotoxins by at least sevenfold compared to the levels in the control. Ferulic acid, sinapic acid, and bromelain even prevented the synthesis of BEA in the strain *F. proliferatum*. Caffeoylquinic acid promoted the accumulation of BEA compared to the control for the *F. ananatum* strain.

The amounts of FB_1_, FB_2_, and BEA produced in FYM (fructose yeast malt), GYAM (glucose yeast asparagine malic acid), and PDA by three *Fusarium* species (*F. proliferatum*, *F. oxysporum*, and *F. ananatum*) are presented in Figure 4. The *F. proliferatum* isolate presented the highest levels of FB_1_ and FB_2_ for all isolates regardless of the medium. The GYAM medium was the most favorable of all the media for fumonisin production. The PJA medium was the most favorable for the production of BEA by the clp081 isolate. Both strains of *F. oxysporum* produced BEA in all media. The PDA medium induced the highest levels of BEA for both *F. oxysporum* and *F. proliferatum* compared to their levels in the other media; the average content of BEA reached 191.3 mg kg⁻^1^ for the clp076 isolate. *F. ananatum* produced more mycotoxins in the FYM medium, a broth that induces fumonisin [55], compared to the other media. Some strains of *F. ananatum* produced both fumonisins and beauvericin. *Talaromyces stollii* did not produce any of our referenced mycotoxins regardless of the medium. Data concerning yeast extract broth (YEPD) were removed from the table due to the absence of detectable mycotoxin content.

### 2.5. Mycotoxins in Pineapple Fruit

The analysis carried out on artificially infected pineapples revealed the in plant production of mycotoxins by *F. oxysporum* and *F. proliferatum* (Figure 5). The black spots with the highest mycotoxin levels were found in the artificially inoculated fruitlets with the mixture of *F. proliferatum* and *F. oxysporum*. Mycotoxin levels reached 0.36 mg kg⁻^1^ FB_1_, 0.12 mg kg⁻^1^ FB_2_, and 0.31 mg kg⁻^1^ BEA, on average, in the infected fruitlets of fruits 1 and 2. The toxin concentrations generally increased between the healthy part of the fruit and the point of inoculation. There were some exceptions though, where concentrations in the different sampling areas were not statistically different. The artificially inoculated strains of *F. ananatum* could not produce mycotoxin in planta. Mycotoxin contamination was also observed in natural fruitlet core rot with 0.01 mg kg⁻^1^ FB_1_, 0.005 mg kg⁻^1^ FB_2_, and 0.02 mg kg⁻^1^ BEA, on average, of the five sampled fruitlets (Figure 6). The fungi present in the naturally infected fruitlets were isolated and cultivated on PDA medium. Morphological identification of the isolates allowed classification of the fungi as belonging to Fusarium and Talaromyces.

## 3. Discussion

The phylogenetic analysis shows that two genera of fungi are the causal agents of fruitlet core rot disease. The Fusarium genus was the most frequently occurring, with 79% of isolates represented by three species: *F. ananatum*, *F. oxysporum*, and *F. proliferatum*. Recent studies have identified *F. ananatum* as a new fungal species responsible for fruitlet core rot. Our results are in agreement with these observations, as *F. ananatum* was the most widespread species in Reunion Island (91% of Fusarium isolates). In fact, *F. verticillioides*, one of the two fungi historically associated with FCR, was not isolated in our collection of 150 strains [24,31,56]. The descriptions of *F. ananatum* are recent, but it is likely to have been the causal agent for many years. For example, an isolate described from pineapple as *F. guttiforme* turned out to be *F. ananatum* (NRRL 2294, CBS 184.29) [16,57]. Based on the phylogenetic analysis of TEF, the clades of *F. guttiforme* and *F. ananatum* are separate, with a high bootstrap value. Our work shows that *F. oxysporum* and *F. proliferatum*, 8% and 1% of Fusarium isolates, respectively, can also produce FCR symptoms. These two fungi have already been isolated in pineapples [17,21], but this is the first time they have been directly related to FCR disease. The four fungi isolated from healthy blossom cups all belong to the genus Fusarium: three were *F. ananatum* and one *F. oxysporum*. They rank on the same clade as their pathogenic counterparts on the phylogenetic tree. Further work on blossom cup isolates could determine the level of contamination by the pathogens and whether the proportion of a particular species is the same between a healthy and an infected area.

This study shows that the Talaromyces genus represents 21% of the isolates collected from diseased pineapple, all of which are *T. stollii*. This species can be distinguished from *T. funiculosus* based on DNA sequence comparisons and morphology. *T. stollii* isolates were supported as being monophyletic, with high bootstrap values. *T. stollii* and *T. funiculosus* are morphologically close. Both species grow fast on general media, with white, green, orange, and red mycelia and the same micromorphological characteristics: biverticillate conidiophores and ellipsoidal conidia. However, *T. stollii* produces colonies that are strongly floccose with more marked red mycelia than *T. funiculosus*, which produces colonies with a stringy funiculose texture. The analysis of ITS, β-tubulin, calmodulin, RPB1, and RPB gene regions was necessary to distinguish *P. funiculosum* from *T. stollii* [58]. *Talaromyces stollii* (CBS 624.93) was previously identified from pineapple without specifying the area of the pineapple from which the strain was isolated [54]. Nonetheless, this work confirms the possibility of finding *T. stollii* in pineapple.

The fungi were collected from pineapples harvested from several producing regions of Reunion. This island offers a wide variability of temperature and rainfall based on its topography. This diversity allows growers to produce pineapples throughout the year. Interestingly, pineapples produced at lower altitudes had a higher Fusarium/Talaromyces contamination ratio than those produced at higher altitudes. Similarly, the Fusarium/Talaromyces ratio was higher in the driest parts of the island. Temperature and humidity are factors that influence the growth of fungi [59]. It appears that Fusarium isolates prefer warm temperatures with little rainfall, and Talaromyces isolates prefer cooler temperatures with more moisture.

These pathogens found on pineapple fruitlets develop when the fruit is ready to be consumed. This timing raises the question of the toxigenic potential of our strains, knowing that both genera can produce a wide range of mycotoxins. The fumonisin B_1_ and B_3_, moniliformin, and beauvericin were monitored in vitro from Fusarium species isolated from pineapple [17]. *Talaromyces stollii* isolates can produce austins and chromophore group “HHH” [54], but this species has never been described as a mycotoxin producer in pineapple. To validate the potential risk of pineapple contamination, mycotoxin contents were measured in naturally and artificially infected fruitlets. Mycotoxins were found in our infected fruit samples. Mycotoxin contamination was also observed in fruits with natural fruitlet core rot, with FB_1_, FB_2_, and BEA occurring at non-negligible levels. Identification of fungi on the naturally infected fruitlets revealed the presence of *Talaromyces stollii* and Fusarium, especially *F. ananatum* and *F. proliferatum*. Higher amounts of mycotoxins were found in inoculated fruitlets compared to naturally infected fruitlets. The time between inoculation and sampling of the infected fruitlets was optimal for producing the highest levels of mycotoxin. We could not predict the stage of the infection in its natural state; mycotoxins may not have been synthesized yet or already degraded. A larger sampling of infected fruit should be carried out to ensure safety of pineapple for consumers. In addition, a pineapple medium (PJA) was developed to establish the mycotoxigenic potential of our isolates under controlled conditions close to natural conditions. The fumonisins B_1_ and B_2_ were detected from twelve strains of *F. ananatum* and *F. proliferatum*. Our results are in agreement with those of Stępień et al. [17], who showed that *F. proliferatum* produced the highest levels of mycotoxins in vitro. *Fusarium oxysporum* produced only beauvericin. Mycotoxin levels were close to the detection limit of the device for *F. ananatum* strains, which may explain why mycotoxin production was detected in a low number of isolates, i.e., this may be underestimated and accurate determination may require more sensitive detection.

Phenolic acids have a variable effect on fungal growth and mycotoxin production, depending on the strain as well as the concentration and type of phenolic acid assayed [60,61,62]. Natural phenolic acids from wheat bran inhibit in vitro trichothecene biosynthesis in *Fusarium culmorum* by repressing *Tri* gene expression [41]. Inhibition of *Fusarium graminearum* growth and mycotoxin production by phenolic extract from *Spirulina* sp. was also observed [63]. Our results show variability in the in vitro accumulation of mycotoxin due to metabolites naturally found in pineapple fruits. The PJA medium was enriched with phenolic compounds or enzymatic complexes at 0.5 g L**^⁻^**^1^, a concentration close to that naturally found in infected fruits [40]. Toxin accumulation was reduced in the presence of each supplemented compound compared to the control for *F. proliferatum.* Only caffeoylquinic acid enhanced the level of BEA for *F. oxysporum*. However, while the contaminating strains are toxigenic in vitro, it seems that their toxigenicity is limited in vivo because of the metabolites produced by the fruit during the infection.

With food and health security being major issues, it is therefore important to produce food with the least possible mycotoxins. In the case of the fruitlet core rot disease, it would be interesting to put non-toxigenic strains in competition with *F. ananatum*. For example, we know that *T. stollii* and *F. ananatum* coexist in the blossom cups. If it is confirmed that *T. stollii* is not toxigenic, its presence could be positive in case it has a detrimental effect on *Fusarium* spp. The result would be even more interesting with non-pathogenic species of fungi naturally present on pineapple.

## 4. Materials and Methods

### 4.1. Determination of Fungal Diversity

#### 4.1.1. Fruit Sampling and Fungal Isolation

Reunion Island (21.1° S, 55.5° E) is located in the Indian Ocean. The climate in Reunion is tropical, but the temperature varies with elevation. Precipitation levels vary greatly within the island, with the east being much wetter than the west. The pineapple crops are spread all over the island. Fifty mature fruits were harvested from 10 farms on the island during the last quarter of 2015. One to eight spots were isolated from naturally infected pineapples. Four healthy looking blossom cups were also sampled. Samples of approximately 6 mm^3^ were deposited on Sabouraud chloramphenicol medium. Daily observations made it possible to transfer the mycelium to potato dextrose agar (PDA) medium. Single conidial cultures were performed after seven days at 25 °C and then stored using a cryopreservation method at −75 °C in 15% glycerol aqueous solution [64]. All isolates used in this study are maintained in the Fusarium culture collection of the CIRAD La Reunion, Ligne Paradis, France, and at the Université de Montpellier, Qualisud, France.

#### 4.1.2. Morphological Comparisons

Preliminary morphological identification work was performed on the pathogens present in the fruits. Isolates were grown on carnation leaf agar (CLA) and PDA for 7 days at 25 °C [65]. Fungal structures produced on these media were mounted on microscope slides and used in the morphological comparison of the Reunion isolates. Colony color was assigned using the color charts of Rayner for isolates grown on PDA for 7 days at 25 °C [66].

Scanning Electron Microscopy allowed the differentiation of fungi genera. Small pieces of fruit in the blossom cup region were observed with a Hitachi S4000 SEM.

#### 4.1.3. DNA Extraction, PCR, Sequencing, and Alignment

The fungi were grown on PDA plates for five days, and genomic DNA was then extracted from fresh aerial mycelia using the Promega Wizard^®^ SV Genomic DNA Purification System (Anatech Instruments (Pty) Ltd., Randburg, South Africa) according to the manufacturer’s recommendations. The extracted DNA was quantified using a Nanodrop spectrophotometer (NanoDrop, Wilmington, NC, USA) and stored at −20 °C until further use.

The primers EF1 (5′-CGAATCTTTGAACGCACATTG-3′) and EF2 (5′-CCGTGTTTCAAGACGGG-3′) were used to amplify a region corresponding to the translation elongation factor-1α (TEF) gene of *Fusarium* sp. isolates [67]. The PCR consisted of 1× Roche Taq reaction buffer with MgCl_2_, dNTPs (250 mM each), primers (0.2 mM each), template DNA (25 ng), and Taq polymerase (0.5 U). The conditions for the amplification of the TEF gene were an initial denaturation at 95 °C for 5 min followed by 30 cycles of denaturing at 94 °C for 45 s, annealing at 58 °C for 45 s, and elongation at 72 °C for 1 min, followed by a final elongation step at 72 °C for 7 min. The resulting PCR amplicons were purified using a QIAquick PCR Purification kit (QIAGEN, Hilden, Germany).

Primers ITS1 (5′-TCCGTAGGTGAACCTGCGG-3′) and ITS4 (5′-TCCTCCGCTTATTGATATGC-3′) were used to amplify the internal transcribed spacer (ITS) region of the remaining strains (White et al. 1990). The PCR assay was carried out in a total volume of 20 μL comprising 1 × KAPASYBR^®^ FAST, 2 μL DNA (10 ng μL**^⁻^**^1^), and 200 nM of each primer. PCR cycling conditions consisted of an initial denaturation step at 94 °C for 5 min followed by 30 cycles at 94 °C for 45 s, 52 °C for 30 s, and 72 °C for 60 s, and a final extension step at 72 °C for 5 min.

Sequencing of both strands was performed using the ABI 3700 DNA Sequencer (Applied Biosystems, Waltham, MA, USA) according to the manufacturer’s instructions in the Genomic Unit of the Universidad Complutense of Madrid (Madrid, Spain). Sequences were edited and aligned using Geneious (Biomatters) [68]. Sequences were deposited in the EMBL database [69], and the accession numbers are shown in Appendix A.

#### 4.1.4. Phylogenetic Analysis

The alignments of TEF and ITS sequences were performed using MUSCLE as implemented in the MEGA7 program [70]. PhyML was used for maximum likelihood (ML) analysis [71]. The robustness of trees in the ML analyses was evaluated by 100 bootstrap replications. The phylogram based on TEF sequences was rooted with *N. solani* (GenBank accession number: DQ247710). *T. dendriticus* (GenBank accession number: JX091486) was selected as the outgroup in the ITS analysis. The output tree files were visualized in FigTree v.1.3.1.

### 4.2. Toxigenic Potential

#### 4.2.1. Pineapple Medium (PJA)

The ability of fungi from the collection to produce mycotoxins was assessed on a pineapple medium optimized for the study. The optimized pineapple juice agar was made from a solution of water agar (5%) mixed with commercial pineapple juice at a ratio of 20/80 with a final agar concentration of 1.25%. Five microliters of a conidial suspension (concentration 10^6^ conidia mL⁻^1^) of each fungal strain was deposited in the middle of the pineapple medium. Petri dishes were incubated for seven days at 25 °C before mycotoxin extraction. One gram of agar corresponding to 3 plugs in the fungal colony was used for mycotoxin extraction. The toxigenic potential of the collection strains was tested on PJA. Two technical repetitions of the liquid chromatography–mass spectrometry (LC–MS) analyses were performed.

#### 4.2.2. Optimal Medium for Mycotoxin Production

Fungal strains of *F. ananatum* (isolates clp028, clp101, clp103, and clp117), *F. oxysporum* (clp076 and clp105), *F. proliferatum* (clp081), and *T. stollii* (clp023 and clp042) were subcultured on solid and liquid media. For PDA and PJA, spore suspensions were deposited on solid media and then incubated at 25 °C for seven days. Three liquid media were tested: YEPD (yeast extract 10 g L⁻^1^, peptone 20 g L⁻^1^, dextrose 20 g L⁻^1^), GYAM (glucose 43 g L⁻^1^, yeast extract 0.5 g L⁻^1^, L-asparagine 1.06 g L⁻^1^, malic acid 0.7 g L⁻^1^, NaCl 0.1 g L⁻^1^, K_2_HPO_4_ 0.766 g L⁻^1^; MgSO_4_ 0.24 g L⁻^1^; CaCL_2_ 1 g L⁻^1^), and FYM (fructose 20 g L⁻^1^, yeast extract 1 g L⁻^1^, malt extract 0.5 g L⁻^1^, peptone 1 g L⁻^1^, KH_2_PO_4_ 1 g L⁻^1^, MgSO_4_·7H_2_O 0.3 g L⁻^1^, KCl 0.3 g L⁻^1^, CuSO_4_·5H_2_O 0.01 g L⁻^1^, ZnSO_4_·7H_2_O 0.05 g L⁻^1^). One mycelium plug was deposited in an Erlenmeyer flask with 50 mL of broth. The liquid media were incubated at 30 °C and stirred at 100 rpm for 10 days. Each modality was repeated four times.

#### 4.2.3. Culture Supplementation with Phenolic Acids and Enzymatic Complex

The isolates clp101 (*F. ananatum*), clp081 (*F. proliferatum*), clp076 (*F. oxysporum*), and clp120 (*T. stollii*) were subcultured on PJA with the addition of secondary metabolites that are commonly observed in pineapple fruits, especially those observed in black spots. Standards of coumaric acid, caffeoylquinic acid, ferulic acid, sinapic acid, and bromelain purchased from Sigma-Aldrich (France) were added at concentrations of 0.25, 0.5, and 1 g L⁻^1^. The control contained only PJA. The compounds were initially dissolved in 0.25% methanol solution to facilitate their dissolution in PJA at 44 °C. The same volume of methanol was added to the controls. Fungal inoculations were made as in previous cultures. After seven days of growth at 25 °C, mycotoxins were extracted, and three independent extractions were performed. Each modality was repeated three times. Two technical repetitions of the LC-MS analyses were performed.

#### 4.2.4. Inoculated and Naturally Infected Fruitlets

Inoculations were performed on mature fruits harvested at the CIRAD Experimental Research Station on Reunion Island. The inoculated fruitlets were originally healthy. Fruitlets were inoculated with 50 µL of a conidial suspension at 10^3^ conidia mL⁻^1^ of isolates of *F. ananatum* (clp101, clp117), *F. oxysporum* (clp076, clp105), *F. proliferatum* (clp081), and *T. stollii* (clp120) using a 0.25 mm diameter needle. Four fruitlets representing the “infected fruitlet” were inoculated on one side of the fruit, whereby two of the fruitlets were inoculated on the upper part while the others were inoculated on the lower part. The syringe was pushed through the peel in the fruitlet perpendicular to the fruit axis. The flesh between infected fruitlets represented the “adjacent fruitlet”. The other side of the fruit represented the ‘healthy fruitlet’. After an eight-day incubation at 24 °C away from direct sunlight, the pineapples were sampled according to the different areas of interest.

Naturally infected fruitlets were obtained from commercially available pineapples originating from Mauritius Island. The inconsistent sample size for healthy and infected fruit is due to the fact that not all pineapples sampled had multiple naturally black spots. Adjacent and healthy areas were also sampled to measure the mycotoxin dispersion in the parenchyma.

#### 4.2.5. Mycotoxin Extraction and Quantification

The extraction of mycotoxins was carried out according to the same protocol for all experiments. Plugs of 1 g of solid media, 10 g of fresh fruit, and 1 g of supernatant and pellet from the broth were used after centrifugation for five minutes (8000 rpm). Samples were mixed with 20 mL of acetonitrile/water/acetic acid (50:50:1). After agitation, salts (modified roQ^TM^ QuEChER EN 15562) were added to allow phase separation. Then, the supernatant was diluted 100-fold with mobile phase A (water + 0.5% acetic acid) and the internal standard mix was added. The HPLC analysis was carried out on a LCMS-8040 Shimadzu (Kyoto, Japan). The separation was performed at 50 °C using a 50 × 2 mm, 2.6 µm particle size, end-capped reversed-phase Phenomenex Kinetex XB^®^. The injection volume was 50 µL, and the flow rate was 0.4 mL min⁻^1^. The mycotoxins were analyzed using the following gradient; 10–55% B (isopropanol + 0.5% acetic acid) in 1.5 min, 55–85% B in 1.5 min, 80–85% B in 0.5 min, and 2–80% B in 7 min, after which the column was washed and equilibrated to the initial conditions. Standards of aflatoxins (AFB_1_, B_2_, G_1_, G_2_), ochratoxin A, fumonisins B_1_ and B_2_, zearalenone (ZEA), trichothecenes (DON, HT-2, nivalenol, 3ADON, 15ADON, DAS, fusarenon-X) (R-Biopharm), and beauvericin (BEA) (Libios) were used to detect and quantify mycotoxins.

### 4.3. Statistical Analysis

All statistical analyses were conducted in R [72]. The graphics were realized with ggplot2 [73].

For the in vitro data, mycotoxin concentrations were estimated with nine independent linear mixed models for each toxin and strain/species combination [74]. The compounds and media were fixed effect. The residuals are independent and their normality was verified graphically.

For the artificial inoculation data, mycotoxin concentrations were estimated with nine independent linear mixed models for each toxin and strain combination. The sampling area (healthy, adjacent, or inoculated) is a fixed effect, whereas the fruit is a random effect (measurements on the same fruit are not independent). Therefore, the residuals are independent and their normality was verified graphically.

In each model, the three sampling areas were compared two by two in a post hoc test following the Benjamini–Hochberg procedure [75,76].

For data concerning naturally infected fruits, the mycotoxin concentrations were estimated using three independent linear mixed models for each toxin. As above, the sampling area (healthy or infected) is a fixed effect and the fruit is a random effect. The residuals are independent and their normality was verified graphically.

## Figures and Tables

**Figure 1 toxins-12-00339-f001:**
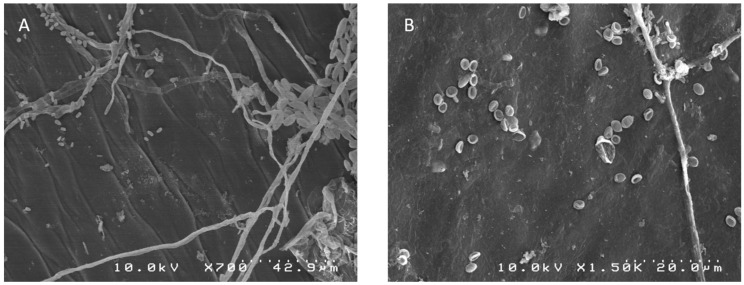
Scanning electron microscopy images of natural fungal colonization of the blossom cup of pineapple fruitlet. (**A**) Represents the mycelial network and spores with a fusiform shape characteristic of *Fusarium* sp. (**B**) Displays mycelium and spores with ellipsoidal shapes belonging to *Talaromyces* sp.

**Figure 2 toxins-12-00339-f002:**
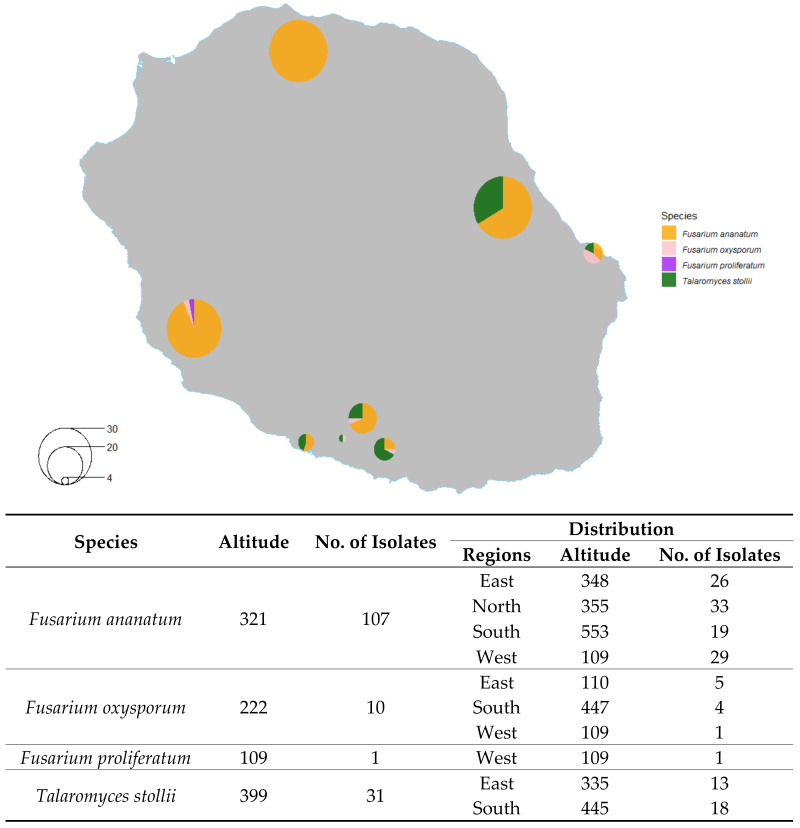
Distribution of isolated fungi species across Reunion Island. The size of the pie chart is proportional to the number of fungi isolated per geographical area.

**Figure 3 toxins-12-00339-f003:**
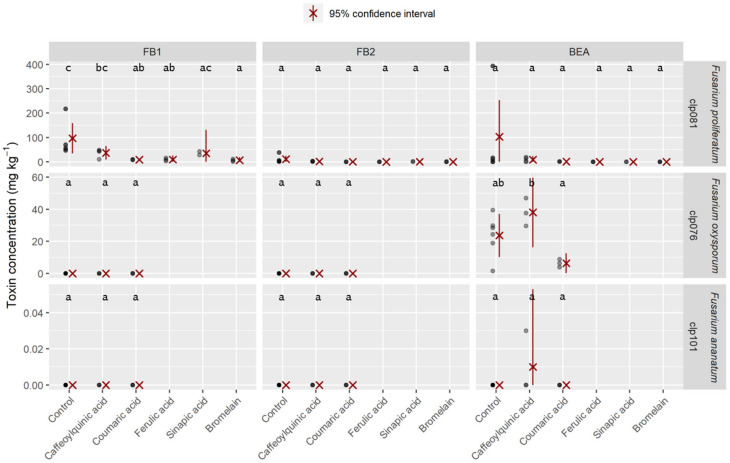
Amounts of fumonisins (FB_1_ and FB_2_) and beauvericin (BEA) produced in PJA medium with the addition of the different compounds (0.5 g L⁻^1^), as indicated, for the growth of different isolates of *Fusarium proliferatum*, *F. oxysporum*, and *F. ananatum.* Different letters indicate that data are significantly different at *p* < 0.05 between compounds (according to Tukey’s multiple comparison test). A gray colored dot is assigned to each measurement, with darker regions indicating overlap.

**Figure 4 toxins-12-00339-f004:**
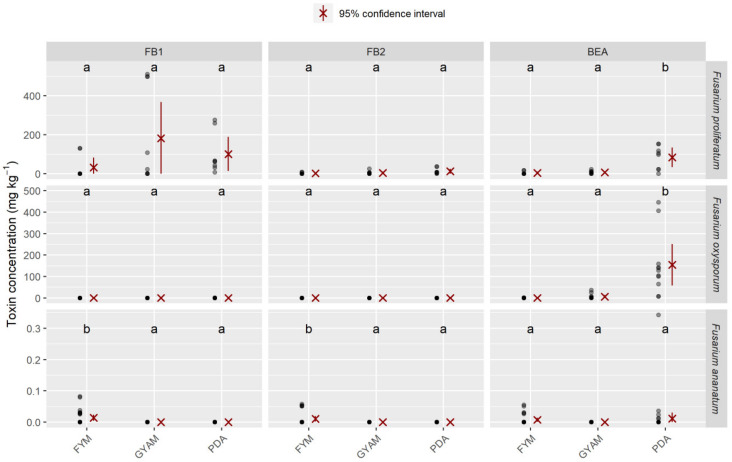
Amounts of fumonisins (FB_1_ and FB_2_) and beauvericin (BEA) produced in different media (fructose yeast malt (FYM), glucose yeast asparagine malic acid (GYAM), and potato dextrose agar (PDA)) for different isolates of *Fusarium proliferatum*, *F. oxysporum*, and *F. ananatum*. Different letters indicate that data are significantly different at *p* < 0.05 between media (according to Tukey’s multiple comparison test). A gray colored dot is assigned to each measurement, with darker regions indicating overlap.

**Figure 5 toxins-12-00339-f005:**
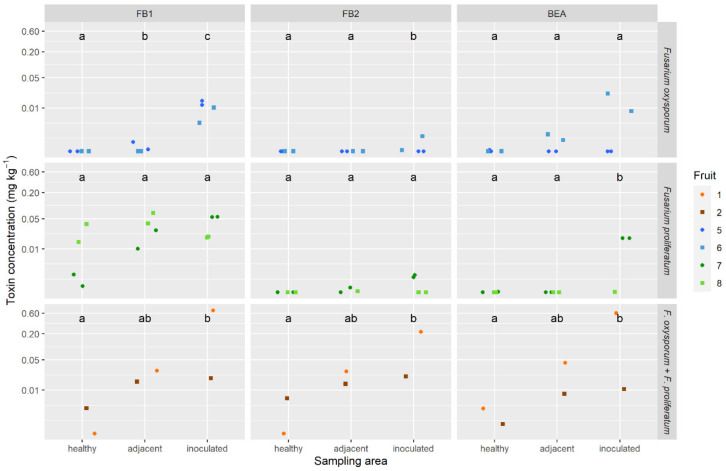
Amounts of fumonisins (FB_1_ and FB_2_) and beauvericin (BEA) produced in the healthy, adjacent, and inoculated fruitlets of pineapple fruits artificially infected with *Fusarium oxysporum* (clp076), *Fusarium proliferatum* (clp081), and a mixture of *F. oxysporum* and *F. proliferatum* (clp105/clp081). Different letters indicate that data are significantly different at *p* < 0.05 between sampled areas (according to Tukey’s multiple comparison test). The different colors of the shapes correspond to a different pineapple fruit.

**Figure 6 toxins-12-00339-f006:**
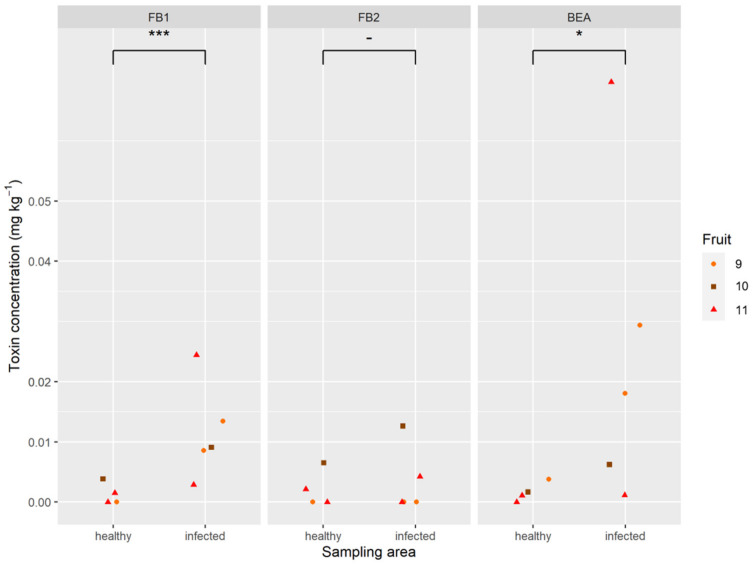
Amounts of fumonisins (FB_1_ and FB_2_) and beauvericin (BEA) produced in healthy and naturally infected fruitlets of pineapple. Differences between sampled areas were either significant at *p* < 0.05 (*), *P* < 0.001 (***), or nonsignificant (-). The different colors of the shapes correspond to a different pineapple fruit.

**Table 1 toxins-12-00339-t001:** Amount of fumonisins (FB_1_ and FB_2_) and beauvericin (BEA) produced in pineapple juice agar (PJA) for mycotoxin-producing isolates.

Species	Collection No.	FB_1_ (mg kg⁻^1^)	FB_2_ (mg kg⁻^1^)	BEA (mg kg⁻^1^)
*Fusarium ananatum*	clp019	0.02	0.01	0
	clp079	0.1	0.24	0
	clp082	0.01	0.02	0
	clp091	0.12	0.24	0
	clp096	0.01	0.03	0
	clp103	0.12	0.24	0
	clp116	0.01	0.05	0
	clp119	0.03	0.03	0
	clp136	0.13	0.13	0
*Fusarium oxysporum*	clp076	0	0	23.68
	clp105	0	0	36.92
*Fusarium proliferatum*	clp081	96.7	11.2	103.2

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
