# Peer review of "Diversity and Toxigenicity of Fungi that Cause Pineapple Fruitlet Core Rot"

_toxins, 2020, doi:10.3390/toxins12050339_

Round 1

Reviewer 1 Report

In this manuscript, the authors investigated the fungal species that caused pineapple fruitlet core rots and their ability to produce mycotoxins under in vitro and in planta conditions. The paper overall is well-constructed, with clear research questions and relevant methods. Sufficient results on the mycotoxin production in vitro and in planta justify their conclusion that limited toxins would be produced in infected pineapple fruitlets because of their metabolites. However, there are a few consistency issues on the figures and tables in this manuscript that should be clarified.

  1. For the result 2.1 morphological identification, providing light microscope photos for these fungal isolates to highlight their structural characteristics would help the readers better understand the methods used to differentiate Fusarium and Talaromyces
  2. Table 1 listed all Fusarium strains isolated in this study, which has too many redundant information. Isolates with the same species, origin and altitude should be compiled together to shorten the table.
  3. In figure 1, pie charts for fungi isolated in separate regions on the island are shown, but the numbers of Fusarium and Talaromyces isolates for each region are not labelled. Providing this data would help to support the later argument on the correlation of fungal distribution and climate.
  4. Please consider fixing the text distance in figure 2 to improve its readability.
  5. For figure 4 and 5, legends should be provided to explain the dot colors. Missing data should be explained (Ferulic acid, Sinapic acid and Bromelain treated clp076 and clp101). Sample size is inconsistent for different treatments, which should also be explained. The confidence intervals should be proofread to confirm their accuracy (For example, Sinapic acid treated clp081, Caffeoylquinic acid treated clp101).
  6. For figure 6, what is the reason for using mixture of clp105/clp081 rather than clp076/clp081? Explanations are needed for taking only single sample for clp081/clp105 treated fruit while two samples on single-strain-treated fruits.
  7. For figure 7, please provide the justification for inconsistent sampling size for healthy and infected fruits.
  8. In part 2.4. Mycotoxins synthesized in vitro, the concentrations used for the five metabolites (Caffeoylquinic acid, Coumaric acid, Ferulic acid, Sinapic acid and Bromelain) are 0.5 g/L. What is the justification for using this concentration and what is the natural concentration in pineapple fruitlets? Providing this information in your discussion would further validate your conclusion.

Author Response

In this manuscript, the authors investigated the fungal species that caused pineapple fruitlet core rots and their ability to produce mycotoxins under in vitro and in planta conditions. The paper overall is well-constructed, with clear research questions and relevant methods. Sufficient results on the mycotoxin production in vitro and in planta justify their conclusion that limited toxins would be produced in infected pineapple fruitlets because of their metabolites. However, there are a few consistency issues on the figures and tables in this manuscript that should be clarified.

  1. For the result 2.1 morphological identification, providing light microscope photos for these fungal isolates to highlight their structural characteristics would help the readers better understand the methods used to differentiate Fusarium and Talaromyces

Unfortunately, access to the microscope is denied due to the Covid19, but I had nice images of SEM. If the structural characteristics of both genera are essential, I will have to wait until the lab reopen to take nice images.

  1. Table 1 listed all Fusarium strains isolated in this study, which has too many redundant information. Isolates with the same species, origin and altitude should be compiled together to shorten the table.

Table 1 and 2 were shorten and compiled together with the map in figure 2.

  1. In figure 1, pie charts for fungi isolated in separate regions on the island are shown, but the numbers of Fusarium and Talaromyces isolates for each region are not labelled. Providing this data would help to support the later argument on the correlation of fungal distribution and climate.

The map has been changed and the pie chart added.

  1. Please consider fixing the text distance in figure 2 to improve its readability.

The text has been updated and the text distance fixed. This figure is now a supplementary material (Figure S1).

  1. For figure 4 and 5, legends should be provided to explain the dot colors. Missing data should be explained (Ferulic acid, Sinapic acid and Bromelain treated clp076 and clp101). Sample size is inconsistent for different treatments, which should also be explained. The confidence intervals should be proofread to confirm their accuracy (For example, Sinapic acid treated clp081, Caffeoylquinic acid treated clp101).

Legends were completed to explain the dot color. A dot grey color is assigned to each measurement, the dot darkens when the concentrations overlap.

Results of statistical analyses were added to the figure.

  1. For figure 6, what is the reason for using mixture of clp105/clp081 rather than clp076/clp081? Explanations are needed for taking only single sample for clp081/clp105 treated fruit while two samples on single-strain-treated fruits.

Experiments were carried out at the end of the project and an allocated budget which made it impossible to redo everything. So we used a strain with the same level of production was the only solution. For the fruits, there were indeed 2 fruits per modality with double inoculation.

  1. For figure 7, please provide the justification for inconsistent sampling size for healthy and infected fruits.

Not all pineapples sampled had naturally multiple black spots. This is the reason why on some fruits there is only one value

  1. In part 2.4. Mycotoxins synthesized in vitro, the concentrations used for the five metabolites (Caffeoylquinic acid, Coumaric acid, Ferulic acid, Sinapic acid and Bromelain) are 0.5 g/L. What is the justification for using this concentration and what is the natural concentration in pineapple fruitlets? Providing this information in your discussion would further validate your conclusion.

The PJA (Pineapple Juice Agar) medium was enriched with phenolic compounds or enzymatic complexes at 0.5 g l-1, a concentration close to that naturally found in infected fruits [41].

Reviewer 2 Report

Comment 1:

Table 1: I suppose NA means not analyzed. However, it should be indicated.

Coment 2:

Page 20, lines 178-179: The addition of caffeoylquinic and coumaric acids to the culture medium did not activate the synthesis of mycotoxins for the F. ananatum tested.

Figure 4 shows that caffeoylquinic acid promotes the accumulation of BEA compared to the control for the F. ananatum strain.

Coment 3:

Figures 4 and 5: They should include statistical analysis, as has been done in Figures 6 and 7.

Coment 4:

Figures 6 and 7: The legends of both graphs are not understood. Do they correspond to different samples taken from different fruits or within the same fruit?

Coment 5:

Page 29, line 363: say « r pm », must say « rpm »

Coment 6:

Page 30, section 4.2.3: The compounds were initially dissolved in 0.25% methanol solution to facilitate their dissolution in PDA.

Was the same volume of methanol added to the controls?

Coment 7:

Page 30, line 378: say «F. oxysporum (clp081), F. proliferatum (clp076, clp105)», must say «F. oxysporum (clp076, clp105), F. proliferatum (clp081)»

Coment 8:

Why is the level of mycotoxins higher in inoculated infected fruitlets than in naturally infected fruitlets?. Authors should include any comments on this topic in the discussion.

Coment 9:

Why do phenolic acids interfere with mycotoxin metabolism? Authors should include any comments on this topic in the discussion.

Author Response

Comment 1:

Table 1: I suppose NA means not analyzed. However, it should be indicated.

Table 1 and 2 were shorten and compiled together with the map in figure 2

Coment 2:

Page 20, lines 178-179: The addition of caffeoylquinic and coumaric acids to the culture medium did not activate the synthesis of mycotoxins for the F. ananatum tested.

Figure 4 shows that caffeoylquinic acid promotes the accumulation of BEA compared to the control for the F. ananatum strain.

Revised as requested

Coment 3:

Figures 4 and 5: They should include statistical analysis, as has been done in Figures 6 and 7.

Results of statistical analyses were added to the figure.

Coment 4:

Figures 6 and 7: The legends of both graphs are not understood. Do they correspond to different samples taken from different fruits or within the same fruit?

They correspond to different samples taken from different fruits. Legends have been changed and the captions too.

Coment 5:

Page 29, line 363: say « r pm », must say « rpm »

Done

Coment 6:

Page 30, section 4.2.3: The compounds were initially dissolved in 0.25% methanol solution to facilitate their dissolution in PDA.

Was the same volume of methanol added to the controls?

Yes, we added it in the text

Coment 7:

Page 30, line 378: say «F. oxysporum (clp081), F. proliferatum (clp076, clp105)», must say «F. oxysporum (clp076, clp105), F. proliferatum (clp081)»

Done

Coment 8:

Why is the level of mycotoxins higher in inoculated infected fruitlets than in naturally infected fruitlets?. Authors should include any comments on this topic in the discussion.

The following comment was added in the discussion: Higher amounts of mycotoxins were found in inoculated fruitlets than in naturally infected fruitlets. The time between inoculation and sampling of the infected fruitlets was optimal to have the highest levels of mycotoxin. We could not predict the stage of the infection in its natural state; mycotoxins may not have been synthesized yet or already degraded.

Coment 9:

Why do phenolic acids interfere with mycotoxin metabolism? Authors should include any comments on this topic in the discussion.

A section has been added on this topic in the discussion and also in the introduction.

Reviewer 3 Report

This paper deals with the identification of fungi associated with the FCR disease of pineapple, and their mycotoxin production. Although no novelties are reported with reference to both aspects, data referred to the specific context of La Reunion, such as the absence of F. verticillioides, can be interesting for comparative considerations with other cropping areas. However, in order to be considered for publication the manuscript requires a comprehensive revision with reference to the following aspects:

1) The citation style must be adapted to the journal's requirements, and correctness of citations must be checked throughout the manuscript, along with their inclusion and format in the final list. In fact, several citations (e.g. Barker 1926, Johansen 1934, Johnson 1935) appear to be missing;

2) Ref. [Nguyen et al 2017] does not report production of patulin by T. funiculosus. Another reference is to be provided to support this correct information, and pertinence of references should be more carefully checked throughout the paper;

3) The introduction section is too diffuse or even verbose; several concepts could be condensed, and repetitions should be avoided;

4) Tables 1 and 2 are definitely unsuitable for publication in the current form. Considering the simple taxonomic structure of the collected sample of isolates, they are merely informative for the location, which actually does not seem to be essential. However, even if authors want to provide this information, they can do it in just a single table reporting how many isolates of each of the 4 species have been collected in each place. The additional information concerning mycotoxin production by Fusarium strains can be provided in a separated table considering only the 10 strains which were characterized in this respect;

5) Again considering the simple taxonomic structure, and the reliability of information derived from TEF sequences, phylogenetic analysis is not at all necessary. It is sufficient to mention % sequence identity with a type strain of each species for a few selected isolates. While the complete list of the sequences obtained for the whole lot can be (eventually) provided as a supplementray file;

6) It is a bit strange that mycotoxin production was detected in a low number of isolates. Authors should comment this finding.

7) The statement that T. stollii did not produce mycotoxins (lines 192-194) must be corrected. In fact, authors can only say that these strains did not produce CERTAIN mycotoxins, among those (WHICH ONES?) they considered as possible products;

8) Statement at lines 208-210 is not clear: do the authors mean that the inoculated fruits were naturally infected with other fungi? If so, how could they discern the respective mycotoxin productions?

9) Several concepts/citations anticipated in the introduction are repeated in the discussion. This redundancy is to be removed;

10) The English language is generally good. However, a further reading by an English native speaker would help adjusting some less fluent parts. Particularly, accordant use of tenses is to be checked throughout the manuscript.

For authors' perusal, I attach an annotated version of the manuscript, which anyway is not exhaustive for the grammar and spelling mistakes.

Finally, a personal consideration concerning ecological relationships between T. stollii and Fusaria: could there be an antagonistic interaction? Particularly, if confirmed that T. stollii is not toxigenic, its presence could be positive in case it have a detrimental effect on Fusarium. I believe that authors should consider such a kind of extension of their work.

Author Response

This paper deals with the identification of fungi associated with the FCR disease of pineapple, and their mycotoxin production. Although no novelties are reported with reference to both aspects, data referred to the specific context of La Reunion, such as the absence of F. verticillioides, can be interesting for comparative considerations with other cropping areas. However, in order to be considered for publication the manuscript requires a comprehensive revision with reference to the following aspects:

1) The citation style must be adapted to the journal's requirements, and correctness of citations must be checked throughout the manuscript, along with their inclusion and format in the final list. In fact, several citations (e.g. Barker 1926, Johansen 1934, Johnson 1935) appear to be missing;

References have been updated for missing citations in the text and adapted to journal requirements

2) Ref. [Nguyen et al 2017] does not report production of patulin by T. funiculosus. Another reference is to be provided to support this correct information, and pertinence of references should be more carefully checked throughout the paper;

Ref. [Nguyen et al 2017] was replaced with two references which report production of patulin by T. funiculosus and a careful check has been carried out for the other references cited:

  1. Vismer, H.F.; Sydenham, E.W.; Schlechter, M.; Brown, N.L.; Hocking, A.D.; Rheeder, J.P.; Marasas, W.F.O. Patulin-producing Penicillium species isolated from naturally infected apples in South Africa. South African Journal of Science 1996, 92, 530-534.
  2. Samson, R.A.; Houbraken, J.; Thrane, U.; Frisvad, J.C.; Andersen, B. Food and indoor fungi; Centraalbureau voor Schimmelculture: Westerdijk Fungal Biodiversity Institute, 2019; Vol. 2, pp. 481.

3) The introduction section is too diffuse or even verbose; several concepts could be condensed, and repetitions should be avoided;

A work has been done to make the introduction more concise. Repetitions were removed in particular thanks to your annotated version of the manuscript.

4) Tables 1 and 2 are definitely unsuitable for publication in the current form. Considering the simple taxonomic structure of the collected sample of isolates, they are merely informative for the location, which actually does not seem to be essential. However, even if authors want to provide this information, they can do it in just a single table reporting how many isolates of each of the 4 species have been collected in each place. The additional information concerning mycotoxin production by Fusarium strains can be provided in a separated table considering only the 10 strains which were characterized in this respect;

Table 1 and 2 were shorten and compiled together with the map in figure 2. The map has been changed and the pie chart added.

The additional information concerning mycotoxin production by Fusarium strains has been provided in a separated table (new Table 1).

5) Again, considering the simple taxonomic structure, and the reliability of information derived from TEF sequences, phylogenetic analysis is not at all necessary. It is sufficient to mention % sequence identity with a type strain of each species for a few selected isolates. While the complete list of the sequences obtained for the whole lot can be (eventually) provided as a supplementary file;

The text has been updated and the text distance fixed in Figure 2 to improve its readability. Those figures are now supplementary materials (Figure S1 and S2). In order to replace the table 1 and 2, % sequence identities with a type strain of each species for a few selected isolates were added in the results.

6) It is a bit strange that mycotoxin production was detected in a low number of isolates. Authors should comment this finding.

The fumonisins B1 and B2 were detected from twelve strains of F. ananatum and F. proliferatum. Fusarium oxysporum produced only beauvericin. Mycotoxin levels being close to the detection limit of the device for the F. ananatum strains may explain that mycotoxin production was detected in a low number of isolates.

7) The statement that T. stollii did not produce mycotoxins (lines 192-194) must be corrected. In fact, authors can only say that these strains did not produce CERTAIN mycotoxins, among those (WHICH ONES?) they considered as possible products;

The text has been replaced as follows: Talaromyces stollii did not produce any of our referenced mycotoxins regardless of the medium (data not shown). The yeast extract broth (YEPD) modality was removed from the table due to the absence of detectable mycotoxin content.

8) Statement at lines 208-210 is not clear: do the authors mean that the inoculated fruits were naturally infected with other fungi? We ensured that the inoculated fruitlets were originally healthy. we added this explanation in the material and method. If so, how could they discern the respective mycotoxin productions?

The fungi present in the natural infected fruitlets were isolated and cultivated on PDA medium. Morphological identifications of the isolates allowed to associate the fungi with the Fusarium and Talaromyces genera.

9) Several concepts/citations anticipated in the introduction are repeated in the discussion. This redundancy is to be removed;

10) The English language is generally good. However, a further reading by an English native speaker would help adjusting some less fluent parts. Particularly, accordant use of tenses is to be checked throughout the manuscript.

The manuscript has undergone professional English editing.

For authors' perusal, I attach an annotated version of the manuscript, which anyway is not exhaustive for the grammar and spelling mistakes.

All your corrections were highly appreciated and incorporated into the revised version

Finally, a personal consideration concerning ecological relationships between T. stollii and Fusaria: could there be an antagonistic interaction? Particularly, if confirmed that T. stollii is not toxigenic, its presence could be positive in case it have a detrimental effect on Fusarium. I believe that authors should consider such a kind of extension of their work.

We asked the same question. This hypothesis is currently being tested in our laboratory with confrontations between different species and mycotoxin analyzes.

Round 2

Reviewer 3 Report

I can see that authors have partly brought the required adjustments. Particularly, although they claim that 'The manuscript has undergone professional English editing' I remark that the English style is still quite approximate throughout the manuscript. Moreover, the reference style is rather scanty, denoting unseemly superficiality (is it so difficult to follow editorial instructions?). Pending the Editor's decision on whether these formal flaws may impair acceptance at this stage, I point out the following additional minor revisions:

line 44: use 'sequencing' instead of 'sequences', since it refers to the method;

line 55: 'reinoculated' instead of 'reintroduced';

lines 58-60: these two sentences should be put together '...to Talaromyces, and Penicillium...';

lines 60-61: correct to: 'T. funiculosus and F. verticillioides were respectively recovered from 67% and 25% of isolations from pineapple';

line 62: delete 'a' at the beginning of the line;

line 69: delete 'potential';

line 74: change to: 'Fusarium oxysporum was found to produce MON and BEA only';

line 109: the word 'phialides' is repeated twice;

line 134: current name of the species Fusarium solani is Neocosmospora solani. This is also to be changed at line 365;

line 192: change to 'Data concerning yeast extract broth (YEPD) were...';

line 209: remove '(data not shown)': if there is no production, there are no data to be shown;

line 212: delete 'natural';

line 214: delete 'genera';

line 231: delete 'clade';

line 234: change 'isolate' to 'species';

line 235 and line 250: delete '(formerly F. moniliforme)' and '(formerly P. funiculosum)': this information has already been provided above;

line 278: delete 'An';

line 366: use the abbreviated form 'T. dendriticus';

lines 373-374: change to 'Five microliters of a conidial suspension (concentration 106 conidia mL−1) of each fungal strain were deposited in the middle of the pineapple medium.';

line 443: change to 'For data concerning naturally...'.

Author Response

Dear Reviewer,

You can find attached the certificate of english editing.

The style of references has been revised to meet the requirements of the journal. I also slightly modified the introduction to make it easier to read.

The additional minor revisions you highlighted have been revised as requested.

Thank you for your involvement.
